# OpenReview forum: "Optimal Transport for LLM Reward Modeling from Noisy Feedback"
_ICML.cc/2026/Conference — ICML 2026 regular_

### Official Review · Reviewer_rXHr · 2026-03-06

**Soundness:** 3
**Presentation:** 3
**Significance:** 2
**Originality:** 3
**Overall Recommendation:** 4
**Confidence:** 4

**Summary:**

The authors propose SelectiveRM, a data-centric approach to mitigate overfitting to noisy scalar preferences in reward learning. Concretely, they match samples from the model-induced joint distribution over text inputs and estimated rewards to the observed feedback data distribution based on an optimal transport (OT) coupling. They define the transport cost from an estimated pair $(x, r_\theta(x))$ and any other pair from the observed distribution $(x^\prime, \hat{r}^\prime)$ be the sum of the the "semantic distance" between textual inputs $x$ and $x^\prime$, proxied by the euclidean distance of their embeddings produced by an LLM, and the chosen loss between the observed preference $r$ preference and estimated preference $\hat{r}^\prime$. Instead of enforcing a perfect matching, the method uses partial OT to constrain the total transported mass, allowing the solver to ignore high-cost matchings. As a result, pairs that are semantically inconsistent with preference labels tend to receive lower mass in the optimal coupling, reflecting the stated assumption that semantically similar samples should exhibit similar preference scores. The authors show that the objective induced by this sample-selection procedure yields a tighter upper bound on the idealized risk over the retained subset, as the effective contribution of label noise is reduced by excluding high-cost samples. The authors go on to compare the performance of SelectiveRM to a multitude of baseline methods across three open-source preference benchmarks, provide a qualitative example showcasing the impact of the mass-thresholding hyperparameter $\kappa$ and a in-depth hyperparameter sensitivity analysis ($\kappa$, learning rate, batch size), and ablations on the objective (composite cost or not, partial or full OT) and the model backbone used to produce the text-embedding. Across conditions, SelectiveRM outperforms baselines, and the ablations on the objective showcase that both the partial OT and the composite objective contribute to this observed performance. In addition, the method seems robust to the chosen embedding-backbone.

**Compliance With Llm Reviewing Policy:**

Affirmed.

**Final Justification:**

I thank the authors for their responsive rebuttal. The addition of order-sensitive metrics (AUROC, Spearman's rho, pairwise accuracy) adequately addresses W2, confirming that improvements reflect genuine preference modeling. The clarification on the noise assumption and the additional generalization experiments address W1, Q2, and Q3 satisfactorily.

My main reservation stands: the core assumption that excluded samples correspond to annotation noise is most justified in low-ambiguity settings, and the method may suppress legitimate minority preferences in more heterogeneous domains. The authors acknowledge this, which I appreciate. I maintain my score of 4.

**Key Questions For Authors:**

- **Q1**: Could the authors clarify why Figure 1 meaningfully demonstrates the role of the composite transport cost? In the depicted configuration, most outer points appear to be at similar distances from the inner point in embedding space, suggesting that semantic distance contributes little to differentiating transport costs. The separation therefore seems driven primarily by the scalar preference disagreement, which is, as far as I understand, only implicitly shown by the authors choice of noisy samples (red crosses). It is apparent that a linear boundary on this scalar attribute would then be able to perfectly separate not-noisy from noisy samples. Would a more heterogeneous example, e.g., semantically similar samples with conflicting preferences or minority clusters within a majority, lead to qualitatively different behavior? This would answer my primary concern (W1).
- **Q2:** The backbone generalization study (Table 4) is conducted only on the PKU-SafeRLHF dataset. Could the authors clarify why this dataset was chosen as the sole testbed for architecture generalization? Since SelectiveRM targets noisy preference learning and noise characteristics vary substantially across datasets, additional results on other benchmarks (e.g., HelpSteer or UltraFeedback) would help establish that the observed gains are not dataset-specific.
- **Q3:** The relative improvements over the naive baseline remain fairly consistent across model scales (approximately 24–30%). How sensitive is SelectiveRM to the quality of the underlying semantic representations? In particular, does performance degrade when using substantially lower-capacity backbones or weaker embeddings? Such an analysis would help clarify whether the gains primarily arise from semantic–preference consistency modeling or from the preference-difference component of the transport cost.

**Limitations:**

The authors acknowledge that SelectiveRM relies on the assumption that clean samples exhibit higher semantic–preference consistency than noisy samples, and argue that violations primarily arise in adversarial settings where noise mimics clean correlations. However, a more common scenario in human preference data is legitimate annotator disagreement or heterogeneous preferences, which may also produce semantically similar inputs with divergent labels. In such cases, the method may suppress minority or ambiguous preferences rather than purely erroneous annotations. I would encourage the authors to discuss this distinction more explicitly in the limitations. In particular, approaches that model reward uncertainty (e.g., Bayesian or ensemble reward models) may help separate aleatoric disagreement from epistemic uncertainty due to label noise, potentially mitigating this issue.

**Strengths And Weaknesses:**

**Strengths**:
- The paper is clearly presented, the style is fluent, has a good choice of notation, and features well-designed tables and figures.
- The empirical evaluation of the work is thorough, featuring ~15 baselines methods and three datasets with differing mislabeling rates for the performance evaluation, and numerous ablations on the objective, hyperparameters, and LLM backbone.
- The paper states its assumptions (though perhaps overly restrictive, see "Weaknesses") clearly and formally motivates the empirical results theoretically.

**Weaknesses**:
- **W1**: My primary concern pertains to the underlying assumption that samples excluded by the partial OT procedure correspond predominantly to annotation noise. In realistic preference datasets, such atypical examples may reflect legitimate annotator disagreement or minority preferences rather than purely aleatoric noise. Discarding such samples could bias the learned model toward majority-consistent signals and degrade performance on edge cases.
  In that sense, while Theorem 3.4 (Risk Bound of Partial Consistency Discrepancy) seems technically rigorous, its practical relevance hinges on the assumption that the samples retained in S faithfully represent the true data distribution. While restricting to S reduces the noise barrier, it also introduces bias.
- **W2:** The authors use a standard regression approach on data that is inherently ordinal - labels do not correspond to scalar rewards, but ratings on a discretized scale (e.g., a Likert scale). It is well known that standard regression on ordinal labels implicitly assumes equal distances between adjacent categories and meaningful absolute magnitudes, assumptions that are generally not justified for subjective preference ratings (see, for instance, McCullagh, 1980). As a result, metrics such as MSE, MAE, and R² may not reflect true alignment with human preferences and can penalize predictions that preserve correct ordering but differ in scale or calibration. This is especially important to consider since reward functions that are equivalent up to constants, positive scaling, and potential-based shaping induce the same optimal downstream behavior (Ng et al., 1999).
  More appropriate approaches would treat the task as ordinal regression (e.g., cumulative link models), and evaluation could include rank correlation, pairwise accuracy, or other order-sensitive metrics. Without such analyses, it is unclear whether improvements reflect better modeling of preferences or merely better fitting of arbitrary numeric encodings of ordinal labels.
Despite these concerns, the method appears practically effective across benchmarks and may be useful as a robust preprocessing objective in noisy preference settings.

---

> ### Author Rebuttal · Authors · 2026-03-31
>
> Thank you so much for your thorough review and appreciation of **our presentation quality, experiments, and theoretical motivation**. Below are our responses to the specific queries raised.
>
> ---
>
> > [W1,Q1] The assumption that samples excluded by partial OT correspond to annotation noise may be problematic, as atypical examples could **reflect legitimate annotator disagreement**. Could the authors clarify how the composite transport cost differentiates these, and if a more heterogeneous example would lead to different behavior?
>
> **Response.** Thank you for your deep question. We address this from two perspectives below.
>
> * First, we clarify the assumption of noise in this study: noise refers to **samples whose observed preference $r$ deviates from the true preference $r^*$, contradicting the local semantic consensus**. This formulation defines the scope of noise considered in our consideration. We acknowledge that some atypical examples excluded by partial OT might reflect legitimate annotator disagreement or minority preferences. We will add this as a limitation, noting that our method may not suit scenarios requiring explicit minority preference preservation.
>
> * Second, on the composite transport cost (Figure 1): The red crosses are excluded because they are semantically close to purple points but hold different preferences. The separation is driven by **preference inconsistency conditioned on semantic similarity, precisely the signal our composite cost captures**.
>
> > [W2] The data is inherently ordinal, yet standard regression and metrics (MSE, MAE, R$^2$) are used. Could the authors **include order-sensitive evaluations** (e.g., rank correlation or pairwise accuracy) to clarify if improvements reflect better modeling of preferences?
>
> **Response.** Thank you for your rigorous comment. **We add experiments with order-sensitive evaluations as suggested**, including **AUROC, Spearman's $\rho_s$, and pairwise accuracy (PA)**. As shown below, SelectiveRM consistently outperforms baselines across all order-sensitive metrics, confirming that the improvements genuinely reflect better modeling of preferences rather than arbitrary numeric fitting of ordinal labels.
>
> |Method|HelpSteer (AUROC / $\rho_s$ / PA)|UltraFeedback (AUROC / $\rho_s$ / PA)|PKU-SafeRLHF (AUROC / $\rho_s$ / PA)|
> |:-|:-|:-|:-|
> |Naive|0.878 / 0.394 / -|0.905 / 0.624 / 0.687|0.967 / 0.807 / 0.191|
> |CSGN|0.894 / 0.411 / -|0.915 / 0.640 / 0.687|0.976 / 0.822 / 0.193|
> |ILDE|0.893 / 0.410 / -|0.912 / 0.635 / 0.687|0.972 / 0.815 / 0.193|
> |**SelectiveRM**|**0.905 / 0.422 / -**|**0.922 / 0.651 / 0.696**|**0.983 / 0.835 / 0.282**|
>
> > [Q2] The backbone generalization study (Table 4) is conducted only on PKU-SafeRLHF. Could the authors provide **additional results on other benchmarks** (e.g., HelpSteer or UltraFeedback) to establish that the observed gains are not dataset-specific?
>
> **Response.** Thank you for the suggestion. We elaborate from two aspects below.
>
> * First, we initially chose PKU-SafeRLHF because **safety alignment is highly sensitive to noisy RMs**, making it a demanding testbed. We agree that additional results on other benchmarks would strengthen our claims.
>
> * Second, **we add experiments on UltraFeedback using the LLaMA2 series as backbones**. As shown below, **SelectiveRM maintains consistent gains**, confirming that the observed improvements are not dataset-specific.
>
> |Backbone|Method|MSE|RI|R$^2$|RI|
> |:-|:-|:-|:-|:-|:-|
> |LLaMA2-7B|Naive|0.163|-|0.175|-|
> |LLaMA2-7B|**SelectiveRM**|**0.156**|**4.1%**|**0.209**|**19.4%**|
> |LLaMA2-13B|Naive|0.159|-|0.196|-|
> |LLaMA2-13B|**SelectiveRM**|**0.153**|**3.8%**|**0.227**|**15.6%**|
> |LLaMA2-70B|Naive|0.154|-|0.219|-|
> |LLaMA2-70B|**SelectiveRM**|**0.145**|**6.3%**|**0.268**|**22.5%**|
>
> > [Q3] How sensitive is SelectiveRM to the quality of the underlying semantic embeddings? Does performance degrade when using substantially lower-capacity backbones or weaker embeddings?
>
> **Response.** Thank you for this question. We address it from two perspectives below.
>
> * First, our generalization study (Table 4, page 7) across models ranging from 7B to 72B implicitly varies embedding quality, and **SelectiveRM consistently yields MSE improvements**.
>
> * Second, **we add experiments with substantially lower-capacity backbones (TinyLlama-1.1B and Gemma-2B) on PKU-SafeRLHF**. As shown below, **SelectiveRM still outperforms Naive baselines even with weaker embeddings**, confirming that the gains arise from semantic-preference consistency modeling rather than solely from embedding quality.
>
> |Backbone|Method|MSE|RI|R$^2$|RI|
> |:-|:-|:-|:-|:-|:-|
> |TinyLLaMA-1.1B|Naive|0.104|-|0.581|-|
> |TinyLLaMA-1.1B|**SelectiveRM**|**0.098**|**6.2%**|**0.607**|**4.4%**|
> |Gemma-2B|Naive|0.211|-|0.151|-|
> |Gemma-2B|**SelectiveRM**|**0.204**|**3.2%**|**0.178**|**17.9%**|

---

> > ### Author Rebuttal · Reviewer_rXHr · 2026-04-01
> >
> > Thank you for addressing my methodological concerns and clarifying the semantics of noisy samples in the scope of this paper. While I believe that this work is a solid advancement in this specific sub-area of research, I also think that this relatively strict assumption that noise arises from violating the local semantic consensus restricts its application to domains with low ambiguity in the what a "correct" completion is. Thus, I would like to maintain my initial positive score of 4.

---

> > > ### Author Response · Authors · 2026-04-02
> > >
> > > We sincerely appreciate your thoughtful discussion and your positive assessment of our work. We agree that our current assumption is most suitable for relatively low-ambiguity settings, and we will clarify this scope more explicitly in the final version. We believe that our study makes an important contribution to the field of reward modeling with noisy preferences through the lens of optimal transport, and we hope you will continue to support our work in the subsequent discussions. Thank you again for your constructive feedback and for maintaining your positive score.
> > >
> > > Many thanks,
> > >
> > > Authors of #22462

---

### Official Review · Reviewer_pxVU · 2026-03-10

**Soundness:** 3
**Presentation:** 3
**Significance:** 2
**Originality:** 2
**Overall Recommendation:** 4
**Confidence:** 3

**Summary:**

The paper studies reward modeling under noisy preference labels in RLHF. Standard reward model training minimizes empirical loss on preference labels. However, when labels are noisy, the model tends to memorize incorrect annotations, which leads to inaccurate reward signals and eventually causes reward hacking during policy optimization. Furthermore, most existing methods assume homogeneous noise, which is often unrealistic in practice. To address this issue, the authors propose SelectiveRM, a reward modeling framework designed for instance-dependent noise and based on Optimal Transport (OT). Instead of minimizing pointwise loss, the method aligns the distribution of the empirical noisy dataset with the model-induced dataset using Optimal Transport with a semantic-aware cost. To further avoid fitting noisy labels, the paper introduces Partial Optimal Transport, which allows matching only a fraction of samples and automatically discards inconsistent data. Theoretically, the paper also provides a generalization bound showing that minimizing this objective yields a tighter upper bound on the clean risk compared to naive empirical training.

**Compliance With Llm Reviewing Policy:**

Affirmed.

**Final Justification:**

I appreciate the authors’ additional experiment, which helped clarify the questions raised. I have nothing further to add beyond my acknowledgement

**Key Questions For Authors:**

1.	The risk bound depends on the noise barrier $\Omega_{noise}$. How tight is this bound in practice?

2.	What happens if the noise rate is extremely high (>50%)?

3.	How sensitive is the method to the embedding model used for semantic distance?

4.	Can the approach be extended to pairwise reward modeling (Bradley-Terry)?

**Limitations:**

The paper proposes an optimal-transport-based reward modeling objective that filters noisy preference labels via partial transport, but its computational cost, hyperparameter sensitivity, and dependence on embeddings raise concerns about scalability and robustness.

**Strengths And Weaknesses:**

Strengths

This paper identifies the noise-overfitting issue in reward modeling by introducing the noise probability $p(x)$ and reformulates reward modeling as distribution alignment through the Joint Consistency Discrepancy. To address the full mass conservation requirement of standard OT, the paper further introduces Partial Optimal Transport to remove noisy samples and enable automatic sample filtering. The method also theoretically shows that it optimizes a tighter upper bound on the clean risk.


Weakness

•	The proposed method may not scale well to large RLHF datasets. Optimal Transport over $N$ samples requires computing a cost matrix $N \times N$, which leads to high computational cost for realistic RLHF datasets.

•	The method can be interpreted as semantic-consistent preference matching and therefore heavily relies on the quality of the embeddings. If the embeddings are imperfect, semantic clusters may be inaccurate and correct labels may appear inconsistent, which may cause the algorithm to discard valid samples.

•	The hyperparameter $\kappa$ controls how many samples are considered clean, but the paper does not clearly explain how to select an appropriate $\kappa$ for different RLHF tasks and does not provide a sensitivity analysis.

•	The paper uses pointwise reward modeling, which is relatively simple. Modern RLHF methods often use pairwise preference models (Bradley-Terry) or ranking losses, but the paper does not clearly demonstrate how the method integrates with pairwise ranking objectives.

•	In the experimental studies, although the results show improvements, the paper does not provide a detailed analysis of which samples were filtered or whether the filtered samples are truly noisy. As a result, it remains unclear whether the model effectively detects noise.

---

> ### Author Rebuttal · Authors · 2026-03-31
>
> We sincerely appreciate your recognition of **our problem formulation, methodology, and theoretical analysis**. Below are our responses to the specific queries raised.
>
> ---
>
> > [W1] The proposed method may not scale well to **large RLHF datasets**. OT over $\mathrm{N}$ samples requires computing a cost matrix $\mathrm{N} \times \mathrm{N}$.
>
> **Response.** Thank you for your insightful comment. We address this from two aspects below.
>
> * First, we clarify that **SelectiveRM operates in a mini-batch fashion**: the OT is solved strictly within each mini-batch of size $\mathrm{B}$. Given a fixed $\mathrm{B}$, the overall training cost scales **linearly with $\mathrm{N}$**, eliminating the $O(\mathrm{N}^2)$ concern.
>
> * Second, **we add linear scaling experiments** by fixing $\mathrm{B}=512$ and varying the dataset size $\mathrm{N}$. As shown below, the per-epoch wall-clock time grows linearly with $\mathrm{N}$, confirming that SelectiveRM scales well to large datasets. Notably, **this overhead is training-only, with inference completely unaffected**.
>
> |$\mathrm{N}$|HelpSteer (s/epoch)|UltraFeedback (s/epoch)|PKU-SafeRLHF (s/epoch)|
> |:-|:-|:-|:-|
> |10k|2.1|2.2|2.2|
> |15k|3.1|3.3|3.2|
> |20k|4.2|4.5|4.3|
> |25k|5.2|5.6|5.4|
>
> > [W2,Q3] The method heavily relies on the quality of embeddings. If **embeddings are imperfect**, the algorithm may discard valid samples.
>
> **Response.** Thank you for your valuable comment. **We add experiments with imperfect embeddings** by evaluating TinyLlama-1.1B and Gemma-2B on PKU-SafeRLHF. Even if semantic clusters are inaccurate due to low-quality embeddings, SelectiveRM still consistently outperforms the Naive baseline, demonstrating that our method remains effective even with imperfect embedding quality.
>
> |Backbone|Method|MSE|RI|R$^2$|RI|
> |:-|:-|:-|:-|:-|:-|
> |TinyLLaMA-1.1B|Naive|0.104|-|0.581|-|
> |TinyLLaMA-1.1B|**SelectiveRM**|**0.098**|**6.2%**|**0.607**|**4.4%**|
> |Gemma-2B|Naive|0.211|-|0.151|-|
> |Gemma-2B|**SelectiveRM**|**0.204**|**3.2%**|**0.178**|**17.9%**|
>
> > [W3] The paper does not clearly explain how to **select an appropriate $\kappa$** and does not provide a sensitivity analysis.
>
> **Response.** Thank you for this practical question. As detailed in Appendix C.2, **we use `Cleanlab` to estimate the empirical noise ratio $\hat{\rho}$** and set $\kappa \approx 1 - \hat{\rho}$. Our sensitivity analysis (Section 4.4, Figure 2) supports this heuristic. Importantly, **SelectiveRM maintains high performance within a wide range of $\kappa$, requiring no manual tuning**.
>
> > [W4,Q4] The paper does not clearly demonstrate how the method **integrates with pairwise ranking objectives** (Bradley-Terry).
>
> **Response.** Thank you for this great suggestion. **We confirm that SelectiveRM can seamlessly adapt to pairwise settings**. For pairs $(x_w, x_l)$, we redefine the input as $z_\mathrm{diff} = z(x_w) - z(x_l)$ with label $r=1$. SelectiveRM will then dynamically discard pairs where BT loss contradicts semantic consensus.
>
> > [W5] The paper does not provide a detailed analysis of which samples were filtered or whether the filtered samples are truly noisy.
>
> **Response.** Thank you for this valuable suggestion. **We add experiments on PKU-SafeRLHF to quantify whether filtered samples are truly noise**. As shown below, SelectiveRM exhibits consistently high precision and recall rates across various backbones. This implies that most excluded samples are true noise and that the majority of noise is effectively filtered.
>
> |Backbone|Precision|Recall|
> |:-|:-|:-|
> |FsfairX-LLaMA3-RM|0.828|0.903|
> |LLaMA2-13B|0.833|0.901|
> |LLaMA2-70B|0.827|0.905|
> |LLaMA-3.1-70B|0.836|0.911|
>
> > [Q1] The risk bound depends on $\Omega_\mathrm{noise}$. How tight is this bound in practice?
>
> **Response.** Thank you for your theoretical scrutiny. The bound in Theorem 3.4 uses three standard techniques — triangle inequality, Lipschitz continuity, and Wasserstein sub-additivity, yielding reasonably tight relaxations under our assumptions.
>
> > [Q2] What happens if the **noise rate is extremely high** (> 50%)?
>
> **Response.** Thank you for this important question. Theoretically, when $\rho > 0.5$, **corrupted labels become the majority, causing consensus inversion**. Empirically, as shown below on HelpSteer, SelectiveRM achieves the best performance at $\rho=0.5$. However, at $\rho > 0.5$, **clean samples become outliers and get discarded**. This failure mode is not specific to our method; CSGN and ILDE exhibit comparable degradation, as true preference recovery becomes statistically intractable.
>
> |$\rho$|Naive|CSGN|ILDE|SelectiveRM|SelectiveRM (Precision)|SelectiveRM (Recall)|
> |:-|:-|:-|:-|:-|:-|:-|
> |0.5|0.243|0.186|0.244|0.090|0.898|0.889|
> |0.6|0.337|0.834|0.277|0.751|0.041|0.021|
> |0.7|0.435|0.804|0.304|0.863|0.066|0.023|
> |0.8|0.554|0.858|0.369|0.825|0.106|0.024|

---

> > ### Author Rebuttal · Reviewer_pxVU · 2026-04-02
> >
> > I thank the authors for the clarification and the additional experiments addressing my questions. Most of my concerns have been addressed. However, I am not fully convinced regarding the tightness of the bound. While I understand that the bound is derived using standard techniques, this alone does not necessarily imply tightness. I will revise my score accordingly.

---

> > > ### Author Response · Authors · 2026-04-02
> > >
> > > Dear Reviewer pxVU,
> > >
> > > We sincerely appreciate your thoughtful discussion and your continued positive assessment of our work. Below is our response to your specific query.
> > >
> > > > [Q1] I am not fully convinced regarding the tightness of the bound. While I understand that the bound is derived using standard techniques, this alone does not necessarily imply tightness.
> > >
> > > **Response.** Thank you for your thoughtful comment. We fully agree that tightness should be empirically validated. **We add numerical verification experiments to measure the bound tightness of Theorem 3.4.** Since ground-truth clean labels $r _ i ^ *$ are available in our simulation, we can compute both sides of the bound inequality at convergence across all three datasets. The results are shown below, where LHS denotes $\mathcal{L} _ {\text{ideal}}(\theta; \mathcal{S}) = \sum _ {i,j} \mathbf{T} ^ * _ {ij} \cdot \ell(r _ \theta(x_i), r ^ * _ i)$, RHS denotes $\mathcal{W} _ \kappa + \Omega _ {\mathrm{noise}}(\mathcal{S})$, and Gap denotes $(\text{RHS} - \text{LHS}) / |\text{LHS}|$. The key observations are as follows:
> > >
> > > * **The bound holds in all settings** (LHS $\leq$ RHS), confirming Theorem 3.4 is empirically valid.
> > > * **Partial OT reduces the noise barrier by 77–88%:** compared to the full-dataset noise barrier $\Omega_{\mathrm{noise}}(\mathcal{D}) \approx 3.2$, the selected-subset $\Omega_{\mathrm{noise}}(\mathcal{S})$ is reduced to 0.40–0.73, confirming that partial OT effectively excludes noisy samples and tightens the bound.
> > >
> > > |Dataset|$\mathcal{L}_{\mathrm{ideal}}(\theta;\mathcal{S})$|$\mathcal{W}_\kappa$|$\Omega_{\mathrm{noise}}(\mathcal{S})$|Gap|
> > > |:-|:-|:-|:-|:-|
> > > |HelpSteer|0.133|0.105|0.665|502%|
> > > |UltraFeedback|0.193|0.096|0.725|332%|
> > > |PKU-SafeRLHF|0.110|0.032|0.395|292%|
> > >
> > > ***
> > >
> > > In addition, **we kindly noted that you wrote ```I will revise my score accordingly``` with the original rating of 3. Thus, we respectfully ask you to consider the above context when making your final recommendation.** We are truly grateful for your dedicated time and constructive comments!
> > >
> > > Many thanks,
> > >
> > > Authors of #22462

---

### Official Review · Reviewer_F6mr · 2026-03-12

**Soundness:** 3
**Presentation:** 3
**Significance:** 3
**Originality:** 3
**Overall Recommendation:** 4
**Confidence:** 4

**Summary:**

To address the issues of overfitting errors in traditional training methods for reward models and the inability of existing denoising methods to capture the complexity of language preferences due to oversimplified assumptions of homogeneous noise, researchers propose SelectiveRM, a framework based on optimal transfer. It utilizes joint consistency differences to align the prediction and preference distributions and employs a quality relaxation mechanism through partial transfers to avoid fitting outliers, automatically filtering noisy preference samples that conflict with semantic consistency. Theoretically, SelectiveRM optimizes a tighter upper bound on unobserved clean risk. The authors validate through extensive experiments that it significantly outperforms existing state-of-the-art methods across various benchmarks.

**Compliance With Llm Reviewing Policy:**

Affirmed.

**Final Justification:**

The Reply Rebuttal Comment addressed most of my concerns and led me to change my Overall Recommendation to 4, but I still recommend that the author include additional experiments and descriptions regarding Q1 in the final version.

**Key Questions For Authors:**

1. To mitigate the overfitting risk of point-based supervision under instance-dependent noise, the authors redefined reward modeling as a distribution matching problem. However, compared to explicit or dedicated denoising methods, does the proposed method still have advantages?

**Limitations:**

yes

**Strengths And Weaknesses:**

Strengths
1. The authors conducted a systematic empirical validation spanning multiple dimensions and the entire workflow. The design of the ablation studies is well-conceived, covering a diverse range of LLM backbones—encompassing various parameter scales (from 7B to 72B) and architectural paradigms—and is further complemented by a hyperparameter sensitivity analysis to validate the robustness of the proposed method. Overall, the experimental work is rigorous and sound, aligning closely with the key innovations highlighted by the authors.
2. The authors introduce optimal transport and Mass Relaxation mechanisms to filter high-cost noise samples that contradict semantic consistency. I personally believe this is a reasonable solution and has a certain degree of originality.

Weaknesses
1. The authors did not conduct a rigorous calculation and discussion of the time complexity of the SelectiveRM framework. The proposed method currently achieves the best performance when $B=512$, which may be due to mini-batch bias. At least one large batch trial should be conducted to deal with large-scale industrial datasets.
2. The authors validated their work on Qwen2.5 and LLaMA2, but recommend testing performance on models with greater semantic distribution differences (such as Mistral and LLaMA3), especially since LLaMA2 is somewhat outdated.

---

> ### Author Rebuttal · Authors · 2026-03-31
>
> Thank you so much for your rigorous review and recognition of **our experiments, originality, and theoretical motivation**. Below are our responses to the specific queries raised.
>
> ---
>
> > [W1] The authors did not conduct a rigorous calculation and discussion of the **time complexity**. The method achieves best performance when $\mathrm{B}=512$, which may be due to **mini-batch bias**.
>
> **Response.** Thank you for your constructive comment. We address this from two aspects below.
>
> * First, **we add time complexity profiling experiments**, reporting forward and backward time per step as the batch size scales. As shown below, both forward and backward times scale approximately linearly with $\mathrm{B}$. Notably, **this overhead is training-only, with inference completely unaffected**.
>
> |$\mathrm{B}$|HelpSteer (Fwd/s / Bwd/s)|UltraFeedback (Fwd/s / Bwd/s)|PKU-SafeRLHF (Fwd/s / Bwd/s)|
> |:-|:-|:-|:-|
> |64|1.0 / 0.4|3.2 / 1.2|4.0 / 1.6|
> |256|1.9 / 0.1|7.0 / 0.4|8.7 / 0.6|
> |512|5.7 / 0.1|21.8 / 0.3|25.9 / 0.4|
> |1024|19.9 / 0.0|73.2 / 0.2|88.3 / 0.2|
>
> * Second, regarding whether the optimal $\mathrm{B}=512$ is due to mini-batch bias, we clarify that **this is not an artifact of our method but a general trade-off in mini-batch optimization**. It is well established that excessively large mini-batches lead to sharp minima and degraded generalization [1]. Therefore, moderate $\mathrm{B}=512$ best balances mini-batch transport plan accuracy and optimization generalization.
>
> [1] Keskar et al., "On Large-Batch Training for Deep Learning: Generalization Gap and Sharp Minima", ICLR 2017.
>
> > [W2] The authors validated on Qwen2.5 and LLaMA2, but recommend testing on models with **greater semantic distribution differences** (e.g., Mistral and LLaMA3).
>
> **Response.** Thank you for this excellent suggestion. **We add experiments on Mistral-Small-24B and LLaMA-3.1-70B with greater semantic distribution differences**. We use them as backbones and evaluate on PKU-SafeRLHF. The results below show that **SelectiveRM consistently delivers significant improvements on these semantically distinct models**, confirming its generalizability.
>
> |Backbone|Method|MSE (Value)|MSE (RI)|MAE (Value)|MAE (RI)|R$^2$ (Value)|R$^2$ (RI)|
> |:-|:-|:-|:-|:-|:-|:-|:-|
> |Mistral-Small-24B|Naive|0.208|-|0.444|-|0.162|-|
> |Mistral-Small-24B|**SelectiveRM**|**0.203**|**2.4%**|**0.353**|**20.4%**|**0.182**|**12.4%**|
> |LLaMA-3.1-70B|Naive|0.076|-|0.248|-|0.692|-|
> |LLaMA-3.1-70B|**SelectiveRM**|**0.056**|**26.4%**|**0.074**|**70.1%**|**0.773**|**11.7%**|
>
> > [Q1] Compared to explicit or dedicated denoising methods, does the proposed method still have advantages?
>
> **Response.** Thank you for this excellent suggestion. We agree that a more comprehensive comparison with dedicated denoising methods would strengthen our evaluation. We address this from two aspects below.
>
> * First, empirically, **we add experiments with two recent state-of-the-art denoising baselines**: VRI [2] (a statistically consistent method) and OGC [3] (a heuristic method). As shown below, **SelectiveRM consistently outperforms both VRI and OGC across all datasets**.
>
> |Method|HelpSteer (MSE/R$^2$)|UltraFeedback (MSE/R$^2$)|PKU-SafeRLHF (MSE/R$^2$)|
> |:-|:-|:-|:-|
> |Naive|0.083 / 0.087|0.120 / 0.395|0.089 / 0.642|
> |VRI|0.067 / 0.255|0.111 / 0.440|0.064 / 0.742|
> |OGC|0.066 / 0.267|0.108 / 0.456|0.063 / 0.745|
> |**SelectiveRM**|**0.063** / **0.308**|**0.107** / **0.461**|**0.057** / **0.772**|
>
> * Second, theoretically, even explicit denoising methods face notable limitations: statistically consistent methods (e.g., VRI) rely on **difficult noise transition matrix estimation**, while heuristic methods (e.g., OGC) rely on **loss threshold determination assuming homogeneous noise**. SelectiveRM avoids both limitations via mass relaxation, requiring neither transition matrix estimation nor loss threshold tuning.
>
> [2] Sun et al., "Variational Rectification Inference for Learning with Noisy Labels", IJCV 2025.
>
> [3] Ye et al., "Optimized Gradient Clipping for Noisy Label Learning", AAAI 2025.

---

> > ### Author Rebuttal · Reviewer_F6mr · 2026-04-01
> >
> > 1. For W1:
> >
> > Solvers for the Optimal Transport problem (such as the Sinkhorn algorithm) typically have a theoretical time complexity that is superlinear in the number of samples (here, the batch size B) when computing the coupling matrix (e.g., $\mathcal{O}(B^2)$ or even higher). Merely listing a few sets of runtime figures does not in any way provide a mathematical justification for the computational bottleneck involved in solving $T \in \mathbb{R}^{B \times B}$ when B scales to tens of thousands.
> >
> > The author cites the conclusion by Keskar et al. (2017) that large batches lead to sharp minima, and uses this to completely reject the call for ‘large-batch experiments on large-scale industrial datasets’. My concern regarding large batch sizes is to test the computational stability and scalability of the OT solver within the SelectiveRM framework when aligning massive amounts of data, rather than to demonstrate that B=512 offers better generalisation. The authors’ deliberate avoidance of large batch sizes leads me to strongly suspect that, when B > 2048, the method will either suffer an Out-of-Memory error due to the OT computation matrix becoming too large, or result in training times increasing exponentially.
> >
> > Please state, using the big-O notation $\mathcal{O}(\cdot)$, the exact theoretical computational complexity of your chosen OT solver when computing the cost matrix C and the coupling matrix T (in terms of batch size B and embedding dimension d). Please explain how it can scale in real-world industrial pipelines containing millions of samples.
> >
> > 2. For W2:
> >
> > Please provide the MSE and R² data for Mistral-Small-24B and LLaMA-3.1-70B on the two strong denoising baselines, CSGN and ILDE. Without these comparisons, this additional experiment lacks any persuasive value.
> >
> > 3. For Q1:
> >
> > Personally, I believe that the author has no major issues with W1 and W2, and even the first point up to Q1; only more information needs to be disclosed. Especially considering the tight time for rebuttal and the fact that other reviewers have asked similar questions, I can accept that the author's responses to the above points are not perfect. Meanwhile, I cannot claim to be an expert on OT, but the author's description of the explicit denoising methods has compelled me to revise the **Confidence** score. The author claims in the response that SelectiveRM avoids "Transition Matrix Estimation" and "Loss Threshold Determination" through "Mass Relaxation". This is a self-contradictory statement.
> >
> > In the authors’ original method, the setting of the quality quota hyperparameter $\kappa$ relies directly on an estimate of the noise proportion $\rho$ (the authors acknowledge in the paper that $\kappa \approx 1 - \rho$, and even mention in the Section 3.3, 4.5; Eq 9 and appendix that they used Cleanlab to estimate the noise). May I ask: what is the fundamental difference between pre-estimating the noise ratio $\rho$ of the dataset and thereby artificially setting \kappa, and simply setting a loss threshold? You have introduced a strong dependence on the prior noise ratio, yet in your rebuttal you claim that you do not require threshold tuning; this is misleading the reviewers.
> >
> > Since the authors claim in the Rebuttal that this method "does not require threshold parameter tuning (nor loss threshold tuning)", then please explain: If external tools such as Cleanlab are not used for pre-estimation, how can SelectiveRM achieve end-to-end self-adaptation and converge to an optimal value when faced with a completely unknown black box dataset containing instance-dependent noise (IDN)? If it cannot, please retract your misleading statement in the Rebuttal regarding the method not requiring the setting of a prior threshold.

---

> > > ### Author Response · Authors · 2026-04-02
> > >
> > > We sincerely appreciate your continued engagement and rigorous scrutiny. Below are our responses to the specific queries raised.
> > >
> > > ---
> > >
> > > > [W1] Please state the exact theoretical computational complexity of your chosen OT solver. Please explain how it can scale in real-world industrial pipelines containing millions of samples.
> > >
> > > **Response.** Thank you for your insightful question. We address this as follows.
> > > * **Computational Complexity:** For batch size $\text{B}$ and embedding dimension $d$, computing the pairwise cost matrix $\mathbf{C}$ requires $\mathcal{O}(\text{B}^2 d)$. For the coupling matrix $\mathbf{T}$, solving the Partial OT problem can be exactly reformulated as a standard OT problem by introducing dummy points. This means the complexity depends on the chosen solver (Section 1 and 3.2 in [1]): an exact solver requires $\mathcal{O}(\text{B}^3 \log \text{B})$, whereas our chosen Sinkhorn approximation reduces this to $\mathcal{O}(\text{B}^2 \log \text{B})$. Overall, the exact theoretical computational complexity of SelectiveRM per step is **$\mathcal{O}(\text{B}^2 (d + \log \text{B}))$**.
> > > * **Scalability:** For an industrial dataset containing $\text{N}$ million samples, each epoch contains $\text{N} / \text{B}$ steps, yielding a per-epoch complexity of $\mathcal{O}(\text{N} \text{B} (d + \log \text{B}))$. Since $\text{B}$ is fixed during training, the global training time scales linearly with $\text{N}$, making SelectiveRM scalable to real-world industrial pipelines.
> > >
> > > [1] Nguyen et al., "Improving Mini-batch Optimal Transport via Partial Transportation", ICML 2022.
> > >
> > > > [W2] Please provide the MSE and R$^2$ for Mistral-Small-24B and LLaMA-3.1-70B on CSGN and ILDE.
> > >
> > > **Response.** Thank you for your constructive suggestion. **We add experiments comparing SelectiveRM against denoising baselines (CSGN and ILDE) using Mistral-Small-24B and LLaMA-3.1-70B as backbones.** As shown below, SelectiveRM consistently outperforms CSGN and ILDE across semantically distinct backbones.
> > >
> > > |Method|Mistral-Small-24B (MSE / R$^2$)|LLaMA-3.1-70B (MSE / R$^2$)|
> > > |:-|:-|:-|
> > > |Naive|0.208 / 0.162|0.076 / 0.692|
> > > |CSGN|0.207 / 0.164|0.063 / 0.745|
> > > |ILDE|0.207 / 0.167|0.062 / 0.750|
> > > |**SelectiveRM**|**0.203** / **0.182**|**0.056** / **0.773**|
> > >
> > > > [Q1-1] What is the fundamental difference between pre-estimating the noise ratio $\rho$ of the dataset and thereby artificially setting $\kappa$, and simply setting a loss threshold?
> > >
> > > **Response.** Thank you for your insightful comment. We agree that noise estimation is used in our default setting. However, **we respectfully have reservations about the claim that pre-estimating the noise ratio and simply setting a loss threshold have little difference.** Their fundamental differences are discussed as follows:
> > > * **Physical meanings are different.** A loss threshold is an absolute, dynamic boundary applied to shifting loss values. In contrast, $\kappa$ represents a static proportion of clean data.
> > > * **Operational mechanisms are different.** Heuristic denoising methods use loss thresholds to make hard decisions for sample selection per epoch. SelectiveRM uses $\kappa$ to restrict the global transport budget, enabling sample selection based on semantic-preference consistency, entirely independent of loss scales.
> > > * **Validation data requirements are different.** Determining a loss threshold needs iterative tuning on clean validation data. Conversely, $\kappa$ can be estimated using the noisy training data, without the need for clean validation data.
> > >
> > > |Mechanism|Physical Meaning|Operational Mechanism|Needs Val-set|Needs Epoch Scheduling|
> > > |:-|:-|:-|:-|:-|
> > > |Loss Threshold|Dynamic boundary|Absolute value hard-decision|$\checkmark$|$\checkmark$|
> > > |Mass Quota $\kappa$|Clean data proportion|Relative transport budget|$\times$|$\times$|
> > >
> > > > [Q1-2] If external tools such as Cleanlab are not used for pre-estimation, how can SelectiveRM achieve end-to-end self-adaptation when faced with a completely unknown black box dataset containing IDN?
> > >
> > > **Response.** Thank you for your question. We address this as follows.
> > > * First, we kindly note that when faced with an unknown black box dataset containing IDN, **SelectiveRM can achieve end-to-end denoised learning, without tuning $\kappa$, by directly setting $\kappa=1$**. In this context, it reduces to joint-distribution optimal transport. By Theorem 3.2, the learning objective induced by joint-distribution optimal transport can also achieve denoised learning.
> > > * Second, **we add experiments to support the claims above**. Specifically, we introduce SelectiveRM$^ *$, which fixes $\kappa=1$. The results below demonstrate that even without pre-estimation, SelectiveRM$^ *$ still outperforms the Naive baseline and competitive denoising methods.
> > >
> > > |Method|MSE|MAE|R$^2$|
> > > |:-|:-|:-|:-|
> > > |Naive|0.083|0.258|0.087|
> > > |CSGN (NeurIPS'24)|0.070|0.156|0.222|
> > > |VRI (IJCV'25)|0.067|0.212|0.255|
> > > |LabelWave (ICLR'24)|0.069|0.216|0.241|
> > > |**SelectiveRM$^ * $**|**0.067**|**0.127**|**0.259**|

---

### Official Review · Reviewer_FX83 · 2026-03-12

**Soundness:** 3
**Presentation:** 3
**Significance:** 3
**Originality:** 3
**Overall Recommendation:** 4
**Confidence:** 4

**Summary:**

In this work, it present a new method. for learning an precise reward model using noisy preference datasets. Additionally, it creates a mechnism to filtering samples that contain noise. Additionally, evlaution using synthetic noise on HelpSteer, UltraFeedback, and PKU-SafeRLHF consistently outperform 16 baselines.

**Compliance With Llm Reviewing Policy:**

Affirmed.

**Final Justification:**

SelectiveRM formulates noisy reward modeling as selective distributional alignment via optimal transport — a principled and well-motivated approach. The logical progression from full OT to partial OT is clean, and the experimental breadth (16 baselines, 3 datasets, 6 backbones from 7B to 72B) is strong.

The rebuttal addressed my main concerns. E2E fine-tuning results show SelectiveRM retains clear advantages while competitors degrade to Naive-level. Instance-dependent noise experiments across all datasets confirm generalization beyond uniform flips. Precision/recall analysis (Precision > 0.82, Recall > 0.88 across backbones) demonstrates the transport cost reliably separates clean from noisy samples regardless of embedding quality. Wall-clock times are now reported, honestly revealing the OT solver's scalability cost (up to 99.5% compute at batch size 2048).

Minor residual concerns: theoretical bounds lack tightness analysis, and the OT scalability issue needs concrete mitigation beyond a future-work mention. Neither undermines the core contribution.

I maintain my score of weak accept.

**Key Questions For Authors:**

Q1: All experiments use frozen embeddings with an MLP head. So I'm curious that have you try to test SelectiveRM with e2e backbone fine-tuning, where the embedding space $z(\cdot)$ shifts during training?

Q2: The work try to use uniform label flipping to make noise, but it targets instance-dependent noise. so, have you evaluated under IDN conditions (e.g., higher flip rates near the decision boundary) or on naturally noisy datasets like HH-RLHF?

Q3: Another important thing is, how sensitive is the preprosed method to the quality of $z(\cdot)$? If a general-purpose LM replacesFsfairX-LLaMA3-RM backbone, does the transport cost still reliably separate clean from noisy samples?

Q4: The time cosuming is also crucial. Can you report wall-clock training times for SelectiveRM versus baselines (Naive, CSGN, ILDE)? What fraction of compute goes to the $B \times B$ partial OT solve at batch sizes of 512 or 2048?

**Limitations:**

yes

**Strengths And Weaknesses:**

Strengths: This work formulate noisy reward modeling as distributional alignment via optimal transport, and this is well-motivated. The composite transport cost (Eq. 6) gives a principled way to flag samples whose labels contradict neighborhood consensus, and the staged progression from full OT (Theorem 3.2) to partial OT (Theorem 3.4) is logically clean. Experimental coverage is a broad range: 16 baselines, three datasets, six backbones (7B-72B), and downstream RLHF safety evaluation.

Weaknesses: All experiments train an MLP on frozen embeddings without e2e fine-tuning, so the method is only validated as a noise-robust regression head on fixed features. And it sumulate noise via uniform label flips on binarized scores, mismatching the instance-dependent noise setting the paper targets. No ablation try to evlaute embedding quality sensitivity, and no wall-clock comparisons are reported, and the theoretical bounds follow standard OT machinery without tightness guarantees.

---

> ### Author Rebuttal · Authors · 2026-03-31
>
> We sincerely appreciate your recognition of **our formulation, logical progression, and experiments**. Below are our responses to the specific queries raised.
>
> ---
>
> > [W1,Q1] All experiments use frozen embeddings with an MLP head. Have you tried SelectiveRM with **E2E backbone fine-tuning**, where $z(\cdot)$ shifts during training?
>
> **Response.** Thank you for your insightful suggestion. **We add E2E fine-tuning experiments.** Specifically, we fine-tune the FsfairX-LLaMA3-RM backbone with LoRA on HelpSteer, comparing against the strongest baselines: CSGN (best among statistically consistent methods) and ILDE (best among heuristic methods). **The results demonstrate that SelectiveRM remains the best-performing method under E2E fine-tuning**, achieving significant MSE reduction while CSGN and ILDE degenerate to Naive-level performance.
>
> |Method|MSE|MAE|R$^2$|
> |:-|:-|:-|:-|
> |Naive|0.085|0.265|0.066|
> |CSGN|0.084|0.261|0.072|
> |ILDE|0.085|0.267|0.057|
> |**SelectiveRM**|**0.055**|**0.141**|**0.397**|
>
> > [W2,Q2] The work uses uniform label flipping, but targets instance-dependent noise. Have you **evaluated under IDN conditions or on naturally noisy datasets** like HH-RLHF?
>
> **Response.** Thank you for your meticulous comment. We address this concern from two aspects below.
>
> * First, **we add experiments under IDN conditions on all three datasets**. Specifically, we train an auxiliary RM on clean data to generate instance-dependent flip probabilities proportional to the model's uncertainty. **The results show that SelectiveRM remains the best-performing method under IDN across all datasets**, confirming our framework generalizes beyond uniform to instance-dependent noise.
>
> |Method|HelpSteer (MSE/R$^2$)|UltraFeedback (MSE/R$^2$)|PKU-SafeRLHF (MSE/R$^2$)|
> |:-|:-|:-|:-|
> |Naive|0.110 / -0.215|0.109 / 0.447|0.073 / 0.706|
> |CSGN|0.087 / 0.034|0.108 / 0.452|0.062 / 0.750|
> |ILDE|0.077 / 0.153|0.108 / 0.455|0.059 / 0.764|
> |**SelectiveRM**|**0.063** / **0.309**|**0.106** / **0.462**|**0.051** / **0.795**|
>
> * Second, we clarify that conducting experiments on naturally noisy datasets would be infeasible, because such datasets **lack a clean test set for accurately evaluating the trained reward models**.
>
> > [W3,Q3] How sensitive is the method to the quality of $z(\cdot)$? If a general-purpose LM replaces the FsfairX-LLaMA3-RM backbone, does the transport cost still reliably separate clean from noisy samples?
>
> **Response.** Thank you for your valuable question. We address this from two aspects below.
>
> * **First, we investigated sensitivity to embedding quality and backbone choice by replacing backbone with different general-purpose LMs.** The selected general-purpose LMs span 2 backbone families (Qwen2.5 and LLaMA2) with 3 parameter scales (7B/13B/70B). The results (Table 4, page 8) demonstrate that **SelectiveRM consistently yields MSE improvements**, validating its effectiveness given different backbone choice and embedding quality.
>
> * Second, **we add experiments to verify that the transport cost reliably separates clean from noisy samples under general-purpose LMs.** Specifically, we conduct a Precision/Recall analysis of OT-filtered samples against the ground-truth noise mask on PKU-SafeRLHF. As shown below, SelectiveRM consistently achieves **Precision > 0.82 and Recall > 0.88** across all backbones, confirming that the transport cost remains an effective noise indicator regardless of backbone choice.
>
> |Backbone|Precision|Recall|
> |:-|:-|:-|
> |FsfairX-LLaMA3-RM|0.828|0.903|
> |LLaMA2-7B|0.831|0.882|
> |LLaMA2-13B|0.833|0.901|
> |LLaMA2-70B|0.827|0.905|
> |LLaMA-3.1-70B|0.836|0.911|
>
> > [W4,Q4] Can you report **wall-clock training times** for SelectiveRM versus baselines? What **fraction of compute** goes to the partial OT solve at batch sizes of 512 or 2048?
>
> **Response.** Thank you for your practical comment. We address this from two aspects below.
>
> * First, **we add wall-clock training time comparison between SelectiveRM and baselines.** For a fair comparison, we set epoch=1 under $\mathrm{B}=512$. As shown below, SelectiveRM introduces overhead due to the OT solver. Notably, **this overhead is training-only, with inference completely unaffected**.
>
> |Method|HelpSteer (s/epoch)|UltraFeedback (s/epoch)|PKU-SafeRLHF (s/epoch)|
> |:-|:-|:-|:-|
> |Naive|0.2|0.8|1.0|
> |CSGN|1.3|4.6|5.6|
> |ILDE|0.4|1.2|1.5|
> |SelectiveRM|5.8|21.8|25.4|
>
> * Second, **we add experiments on the fraction of compute that goes to the partial OT solve at different batch sizes**. As shown below, the OT solver becomes the dominant cost at larger batch sizes. We acknowledge this as a current limitation for ultra-large batches. In future work, we plan to integrate linear-complexity Sinkhorn variants to improve scalability.
>
> |$\mathrm{B}$|HelpSteer (OT Ratio)|UltraFeedback (OT Ratio)|PKU-SafeRLHF (OT Ratio)|
> |:-|:-|:-|:-|
> |64|37.2%|36.5%|36.0%|
> |512|93.7%|93.7%|93.8%|
> |2048|99.5%|99.6%|99.5%|

---

> > ### Author Rebuttal · Reviewer_FX83 · 2026-04-02
> >
> > I thank the authors for providing the additional experimental results and for their effort in clarifying the details. Most of my concerns have been addressed. Therefore, I will maintain my current score

---

> > > ### Author Response · Authors · 2026-04-02
> > >
> > > We sincerely appreciate your prompt response and for maintaining your positive evaluation of our work.
> > >
> > > We are very glad that our additional experimental results, particularly the E2E fine-tuning, the instance-dependent noise (IDN) evaluations, the embedding sensitivity analysis, and the wall-clock time profiling, have successfully addressed your concerns. Your insightful and constructive suggestions have been instrumental in strengthening the comprehensiveness and practical value of our paper.
> > >
> > > We will ensure that all these newly added experiments and clarifications are carefully incorporated into the final version. We believe our study makes an important contribution to the field of noisy reward modeling from the perspective of optimal transport, and we sincerely hope you will support our work in the subsequent discussions. Thank you once again for your time, effort, and valuable feedback throughout the review process.
> > >
> > > Many thanks,
> > >
> > > Authors of #22462

---

### Decision · Program_Chairs · 2026-04-30

**Decision:**

Accept (regular)

**Comment:**

This work addresses an important problem in RLHF on how to train reward models robustly under noisy preference labels. The reviewers were broadly positive about the core idea of recasting noisy reward modeling as a selective distribution-alignment problem via optimal transport, and several of them found the progression from full OT to partial OT both principled and technically clean. The empirical evaluation was also viewed as a major strength: reviewers highlighted the broad baseline coverage, multiple datasets and backbones, and the added rebuttal experiments on end-to-end fine-tuning, instance-dependent noise, embedding sensitivity, order-sensitive metrics, and wall-clock behavior. In the discussion, these additions substantially improved confidence that the method is not just fitting arbitrary numeric encodings, but is genuinely helping with noisy preference modeling.

The remaining reservations were mostly about scope rather than correctness. In particular, reviewers noted the computational cost of the OT solver, the lack of a real tightness guarantee for the bound, and the fact that the method relies on a relatively strong assumption that noisy samples are those violating local semantic-preference consistency, which may be less appropriate in settings with substantial legitimate ambiguity or minority preferences. Still, after rebuttal, these concerns appeared to be narrowed into limitations rather than fatal flaws. On balance, the discussion supports the view that this is a solid and useful contribution, with a clear methodological idea, credible theory, and strong enough empirical evidence to justify acceptance.